# Negative Aspects of Self-Imposed Evacuation among Mothers of Small Children Following Japan’s Fukushima Daiichi Nuclear Power Station Accident

**DOI:** 10.3390/ijerph21050592

**Published:** 2024-05-04

**Authors:** Hitomi Matsunaga

**Affiliations:** Department of Global Health, Medicine and Welfare, Atomic Bomb Disease Institute, Nagasaki University, 1-12-4 Sakamoto, Nagasaki 8528523, Japan; hmatsu@nagasaki-u.ac.jp

**Keywords:** Fukushima Daiichi Nuclear Power Station Accident, self-imposed evacuation/voluntary evacuation, mother

## Abstract

This study clarified the negative aspects of the self-imposed evacuation of mothers of small children seeking to avoid radiation exposure from the Fukushima Daiichi Nuclear Power Station accident on 11 March 2011. We conducted semi-structured interviews with 27 mothers, employing open-ended inquiries based on an interview guide. Our analysis of their responses using the Ka-Wakita-Jiro (KJ) method categorized the results into eight distinct groups comprising 142 labels. These categories included continued anxiety about the health effects of radiation, differences in risk perception, changes in spousal relationships, the inability to make friends and find support, living as a single parent, financial concerns, the unfamiliar feel of the area to which they evacuated, and uncertainty about the future. Despite their hardships, the mothers continued their self-imposed evacuation to avoid radioactivity. Our findings underscore that their anxieties about radiation exposure persisted even after self-imposed evacuation, leading to deteriorated relationships with key individuals who would have been involved in raising their children. These results offer valuable insights into the challenges experienced by the indirect victims of the nuclear accident, such as the mothers of small children.

## 1. Introduction

The Tokyo Electric Power Company’s Fukushima Daiichi Nuclear Power Station (FDNPS) accident occurred on 11 March 2011, in eastern Japan, on the Pacific coast. During the accident, radionuclides were released into the atmosphere and deposited throughout the environment, according to the weather conditions at the time of the venting and hydrogen exposure [1]. The radionuclides expanded to 200 km from the FDNPS, for example, in Japan’s main metropolitan areas in Tokyo and Chiba, etc. However, based on scientific evidence, it was clear that the health effects of external exposure were negligible for people living even near the FDNPS. Furthermore, internal exposures were controlled, thanks to prompt restrictions on food regulations to address the contamination [2]. All international organizations focused on radiation, such as the International Atomic Energy Agency, the United Nations Scientific Committee on the Effects of Atomic Radiation, and the International Commission on Radiological Protection, stated that radiation health effects, including those in children and offspring effects, were not anticipated from direct exposure following the accident in Fukushima [3,4]. However, the release of radionuclides and radiation anxiety led to complex social problems. One of these was self-imposed evacuation (referred to as self-evacuation). 

It has been reported that self-evacuations were carried out not only in the immediate aftermath but also up to several months after the FDNPS accident, and they may still be ongoing [5]. A large number of people experienced anxiety about the unknown effects of radiation exposure from the nuclear accident, which was further exacerbated by unfounded social media messages and malicious rumors containing misinformation, leading to accelerated self-evacuation [6,7]. Among those evacuated by the FDNPS, it was clarified that the people who used public relations information from the local government had lower levels of health anxiety [8]. As a result of this heightened sense of fear, it was estimated that more than 400,000 people self-evacuated, despite only 41,000 individuals who lived near the FDNPS being placed under a formal evacuation order [9]. It is noteworthy that most of the self-evacuees were mothers and small children, due to the increase in thyroid cancers among children as a result of exposure from the Chornobyl accident in 1986 [10]. The younger female population generally experiences greater levels of anxiety regarding radiation exposure compared to male and older people [11]. Additionally, past research has shown that numerous mothers and small children have evacuated from areas deemed safe, not only in the Fukushima Prefecture but also in other areas [12]. Despite this specific situation, there have been a limited number of studies released to date regarding these self-evacuees.

The self-evacuees undertook a self-determined evacuation; therefore, they were not eligible for the support of the Japanese government. Some lucky people found some support from local authorities, volunteer groups, or individuals and have relied on them during their self-evacuation. However, in many cases, they did everything by themselves [13]. In the arrangement of evacuation order areas, including management investigations, the government, and local authorities’, standards for supporting evacuees were established based on international organization recommendations for radiation doses [14]. Areas of self-evacuation had exposure doses that were considerably below those of the Emergency Exposure Situations regulations from the ICRP concerning radiation exposure [1]. However, the mothers of children evacuated from an area without instructions and continued to live as self-evacuees.

The difficulties and concerns of the self-evacuees were clarified by the fathers of the families who had chosen to evacuate voluntarily due to the FDNPS accident. It was reported that the fathers experienced a decline in both their physical and mental health because of the self-evacuation in their families [15]. Further, the process of self-evacuation in mothers with small children was clarified in a previous study. It was clarified that they experienced high anxiety about radiation exposure and the radiation health effects for their children. The motivation behind their decision to self-evacuate clearly indicated radiation anxiety from the FDNPS accident [16]. Nevertheless, the difficulties of the continued self-evacuation of mothers with small children have not been clarified and systematically organized.

The present study clarified the negative aspects of self-evacuation experienced by the mothers of small children, as revealed through interviews with them. The findings contribute to a better understanding of these women and provide valuable insights for establishing support structures based on the negative aspects of self-evacuation in the event of future unexpected nuclear accidents.

## 2. Materials and Methods

### 2.1. Study Participants

The study comprised 27 mothers who, between June 2014 and July 2016, evacuated to Kyushu Island with their children aged 0–6 years, following the FDNPS accident. Local government administrative departments and two volunteer groups in Kyushu facilitated participant recruitment, with participants being recruited via a written explanation of the study provided by the head or leaders of each group (snowball sampling). All procedures for prospective participants were conducted by the researcher.

Each participant lived in a household with only one mother and one to three children, and two women were divorced at the time of the interviews. The areas from which the participants had evacuated were located near Fukushima Prefecture or Tokyo, including Yamagata (1), Fukushima (5), Ibaraki (1), Gunma (2), Tokyo (8), Chiba (8), and Kanagawa (2). The participants ranged in age from their 20s to their 40s, with one in her 20s, six in their 30s, and twenty in their 40s. No participants reported that relatives or friends were killed or injured due to the disaster.

### 2.2. Data Collection

For data collection, we used semi-structured interviews that included open-ended questions based on an interview guide. The interview guide was designed to gather information on the mothers’ difficult experiences as they continued their voluntary evacuation with their children. We asked our participants about any changes or differences in their lives before and after their evacuation. An author conducted the interviews and had no contact with the interviewees before or after the study. The interview guide was pilot-tested with a university student who had children and was reviewed by a senior researcher in qualitative methodologies. All participants were informed about the confidentiality and consent process, and the interviews were recorded, with their consent, using a digital voice recorder. Afterward, the recordings were transcribed verbatim and analyzed. The median length of the interviews was 120 min (range: 106–187 min). We found that after 27 interviews, most of the data points had already been discussed, so we declared that data saturation had been reached.

### 2.3. Analysis

The study data were analyzed using the KJ method, which was developed by cultural anthropologist Jiro Kawakita in 1967 [17]. This method classifies data by emphasizing similarities, even when they may seem different at first glance. The analysis process and structure were as follows: (1) Label creation—verbatim transcripts were carefully read, and labels were created to indicate experiences of difficulty related to the evacuation. (2) Group creation—similar labels were grouped by these difficulties based on participants’ lives since the evacuation, and the groups were checked for similarity. (3) Nameplate creation—the core concepts of all labels were abstracted into a single group, and a nameplate was made for each group. The process of creating nameplates for all groups constituted the end of one level. This process was repeated across several levels until there were no more than 10 nameplates in all. (4) Spatial arrangement—nameplates were compared, and configurations of interrelationships that were easy to understand were arranged. (5) Affinity diagram—the locations of the labels were fixed, and they were arranged at each level like islands. The interrelationships among islands were then examined to draw an overall relationship diagram.

### 2.4. Ethics

This study was approved by the ethics committee of Saga University Graduate School of Biomedical Sciences (No. 19083003). Written consent was provided by participants following their receipt of written and oral explanations from the researcher. Participants’ personal information was protected, and participants were assured that their refusal to participate would not disadvantage them; that all information obtained would not be used for any purpose unrelated to the study; and that participants were free to withdraw their consent or discontinue their participation at any time, and that doing so would not disadvantage them.

## 3. Results

According to the KJ method, the difficulties related to self-evacuations with children were categorized into eight distinct groups, encompassing a total of 142 labels, as illustrated in Figure 1. This classification system was utilized to elucidate the challenges faced by these mothers. The following provides a summary of the most representative narrative for each group.

Continued anxiety about the health effects of radiation; Differences in risk perception; Changes in spousal relationships; Inability to make friends and find support;Living as a single parent;Financial concerns;Unfamiliar feel of the area to which they evacuated; Uncertainty about the future.

### 3.1. Continued Anxiety about the Health Effects of Radiation 

Mothers opted to evacuate to protect their children’s health against radiation exposure following the nuclear accident. Therefore, they selected an evacuation destination that was as far as possible from the FDNPS, even if it meant settling in an unfamiliar region with no family ties. They believed that this area would have the lowest radiation exposure levels in Japan. However, they encountered information on social media that falsely claimed that food, water, and daily necessities distributed throughout Japan were contaminated with radionuclides. Consequently, they continued to feel anxious about radiation exposure and avoided foods and basic necessities made near the FDNPS, deeming them harmful to their children. These avoidance behaviors negatively impacted their relationships with their spouses, relatives, friends, and residents in both the disaster-stricken and evacuation destination areas (Figure 2).


*“Even after evacuating, my anxiety about radiation exposure did not disappear. I am very disappointed that my family and friends think that I am a crazy person because of avoid foods from around Fukushima. However, I am incredibly frightened that my child and I will be exposed if we eat them, so I cannot do it. Everyone around me did not understand my anxiety, so my children and I gradually became isolated.”*


### 3.2. Differences in Risk Perception

The mothers who were interviewed had a clear understanding that they held different perceptions of the risks posed by the radiation from the FDNPS incident compared to those around them. Nevertheless, they believed in the adverse effects of radiation on their children that they had learned about from a workshop seminar and online resources, including from professionals such as radioactivity researchers and medical doctors. These differing perceptions of radiation risk led to strained relationships with key individuals who could have provided support in raising their children, such as their spouses, friends, and local community members. Consequently, they experienced a growing sense of social isolation. 


*“I have gained a lot of knowledge about radioactivity being harmful to our children from various professionals. The Japanese government is allegedly concealing the truth to avoid confusion among the general public. Therefore, I deeply trust honest, truthful information, even if it is not the majority opinion.”*


### 3.3. Changes in Spousal Relationships 

Following the FDNPS accident, these mothers felt sorry for their husbands. The majority of the study participants continued their self-evacuation, despite the lack of understanding from their partners. Their husbands believed that no adverse health effects from radiation would occur in the areas where they resided, and, thus, they desired their wives and children to return home as soon as possible. Conversely, the mothers wanted to continue the evacuation as long as possible, to avoid unnecessary exposure. Differences in their risk perceptions prevented them from being together and weakened their ties as spouses.


*“When I informed him about my concerns regarding self-evacuation, he urged, “Promptly return to our home.” He could not understand that we should protect our children from the evil exposure. I wanted to continue evacuation life as long as possible, so I decided not to argue with him. Consequently, I could not rely on him and I avoided communicating with him increasingly.”*


### 3.4. Inability to Make Friends and Find Support

In addition to the above, the mothers sensed differences in their risk perceptions compared to those of the local residents of the area to which they had evacuated. Almost no local people were concerned about radiation effects from the FDNPS accident. They could not understand why these mothers had self-evacuated with their children to an area previously unknown to them. These women’s differences in risk perception were also noteworthy in comparison to other evacuees. In particular, the evacuees from Fukushima received thyroid checks and other forms of support from the Japanese government, but those who opted to self-evacuate did not. These self-evacuees faced tough experiences, as no one understood their anxiety or recognized them as true evacuees.


*“When I attempted to communicate my strong desire to avoid even minimal radiation exposure, others considered me an eccentric and foolish individual. Moreover, they avoided me, leaving me unable to converse with anyone, even those who expressed a willingness to assist me and my child. I consistently rejected their support, fearing that I would be subjected to harassment. Especially, I was so scared that my child would be harassed.”*


### 3.5. Living as a Single Parent

Living separately from essential individuals who contribute to raising their children, like spouses and family members, these mothers lived life as single parents. Some children were unable to adapt to the sudden changes in their home environment and showed symptoms of stress. Despite this additional weight, these mothers refused to stop their evacuation and return because of their anxiety about unnecessary radiation exposure. Furthermore, the mothers strongly felt a responsibility to raise their children safely, therefore they did not have peace of mind.


*“I am always feeling a strain on my nerves because I have all the responsibility of raising my children. No one helps me take care of my child. Furthermore, everything involved in household chores, such as cleaning, washing, shopping, and cooking, is also up to me. I just want to protect the health of my child from radiation, although I feel that could not enough care for my children by myself. We still keep married, but I feel like I am in a similar state to a single parent.”*


### 3.6. Financial Concerns

Those who self-evacuated were not afforded financial compensation by the government, unlike those who were ordered to evacuate. However, this resulted in a doubling of their family’s cost of living, as they were required to pay for groceries, rent, and utilities in both their original home and new location, and were unable to find new employment due to having small children. These financial hardships led to a deterioration of their relationships with their spouses. 


*“The main cause of my quarrels with my husband was financial problems. He said, ‘If you cannot get along without money from me, come back home immediately.’ I did not understand why he would not cooperate with protecting our children from the radiation exposure from the FDNPS accident.”*


### 3.7. Unfamiliar Feel of the Area to Which They Evacuated 

The primary objective of self-evacuation was to minimize their exposure to the artificial radionuclides released from the FDNPS accident. Typically, the mothers chose their evacuation destinations based on their distance from FDNPS, aiming to be as far away as possible. Consequently, they lacked familial or social connections in the new area. The local people did not share common topics of conversation because they displayed little interest in the FDNPS accident and its radiation-induced health consequences. Further, they were unable to understand the strong accents, characteristic words, and unique culture of the rural areas to which they evacuated. 


*“I could not understand what local people wanted to say to me because they used unfamiliar words and accents. I was very surprised that the language was so incredibly different, even though we were all in Japan. They lived a “normal life,” though the area also had a lot of risks and dangers from radiation exposure. I could not believe that most people were not interested in the evil effects of the FDNPS. The local people were incredibly different from me.”*


### 3.8. Uncertainty about the Future

These mother’s husbands continued working in their hometowns, so they could not quit their jobs and join their families in evacuating. All the husbands desired the return of their wives and children. Conversely, the mothers were reluctant to leave their new area, even though it meant living separately from their spouses. Further, they aimed to protect their children from reputational damage, with some expressing concern that their offspring might have babies with radiation-related health issues from the FDNPS. These mothers did not engage in constructive discussions with their husbands on this matter, because they wished to prolong their evacuation for as long as possible.


*“My biggest concern was how long my children could continue to self-evacuate. I want to continue the evacuation life as long as possible. Because I want to avoid harmful rumors when it comes time for their marriage, or when they have children of their own. I think that some people think radiation will have health effects on the offspring of children who lived through the disaster. However, my husband disagrees with continuing the evacuation. Therefore, the children and I do not have a well-grounded life.”*


## 4. Discussion

This research focused on the negative aspects of self-evacuation for 27 mothers of young children who evacuated following the FDNPS accident. The eight category labels were organized into hard experiences and relationships with key persons for the raising of their children, following the KJ method. The mothers had strong feelings of anxiety about radiation exposure from the artificial radionuclides released during the accident. The most commonly noted finding was that their anxiety about radiation exposure did not disappear, even after they evacuated. Consequently, they could not end their self-evacuation, even when they were in miserable situations. 

Another key finding of the study was that these women’s negative experiences of self-evacuation had come from their radiation risk perception. The “risk perception” element of the study revealed that they feared radiation health effects would occur because of the FDNPS accident. Furthermore, they deeply trusted the information they had personally gathered on this topic. Their risk perception differed from their husbands and others, leading to struggles and negative experiences such as changes in their spousal relationships, an inability to make friends and find support, financial concerns, the unfamiliar feel of the area to which they evacuated, and raising their children as a single parent. In the end, they reported feeling continuous uncertainty about the future. As these mothers had imposed their evacuations on themselves, they could also choose to end them based on their own decisions at any time. However, their negative risk perception about radiation and related anxiety were the primary reasons for their prolonged evacuation.

Like the people who were ordered to evacuate, they decided not to return to their hometowns because of radiation anxiety [18]. It is essential to respect the choices of all evacuees, whether they wish to continue living in their new locations or return to their original homes. However, as this study has revealed, self-evacuation with small children leads to significant struggles. The mothers who shared their experiences with us faced financial struggles, changes in spousal relationships, and the challenge of raising their children as single parents; further, they felt socially isolated in their new communities. These situations of social isolation and poverty are well-known risk factors for child abuse and neglect [19]. This study has highlighted the essential need for comprehensive and flexible support for small children and their mothers, including self-evacuees. In cases of complex situations where it is difficult to address the root of the problem, it is essential to provide flexible support for the children involved for the foreseeable future.

These mothers strongly believed in the harmful health effects of radiation from the FDNPS accident, especially for children. However, scientific evidence from environmental monitoring and past findings from the atomic bombings at Hiroshima and Nagasaki [20] indicates that there are no anticipated radiation health effects stemming directly from the accident. This is an essential point of view officially recognized by international organizations regarding radiation [21]. Nevertheless, the mothers held opposing views on radiation and the FDNPS accident. A previous study clearly showed that people whose opinions opposed those officially recognized by international organizations had often gathered information from social media and other internet sources. These sources claimed that radioactivity would cause serious and various health problems for children [22,23]. Additionally, some individuals believe that the nosebleeds, coughs, and skin symptoms that appeared after the FDNPP accident were caused by radiation exposure. It is challenging to change people’s negative risk perceptions related to radiation when they are based on radiation anxiety. Moreover, it is almost impossible to change risk perceptions that are rooted in a distrust of the government or local organizations [24,25]. 

The mothers who chose self-evacuation just wanted to protect their children from artificial radiation exposure. We should understand and respect their concerns. The solutions to complex issues such as radiation are based on trust, so communicators with these people need to accept their values and thinking. Therefore, it is essential to listen to their opinions and fears and build their trust [26]. Of course, communication about radiation should be consistently based on scientific evidence. After the FDNPS accident, recognition of the importance of information literacy education has been increasing [27]. This study emphasizes the significance of providing information and education about the health effects of radiation to mothers and children particularly. The study findings indicate that such education should not be restricted to the areas near nuclear power plants, but should be extended to all members of society. In Japan, it is common for schools to conduct fire or natural disaster drills annually for children. It is recommended that radiation education be incorporated into the existing curricula for children in schools. Additionally, it is recommended that small meetings and simple presentations be used to provide information about the impact of radiation on pregnant mothers and young children to new parents.

Finally, one of the most serious issues related to these self-evacuees was the dignity of their children. The mothers had a strong urge to protect them, not only in terms of their health but also against discrimination stemming from any association with the nuclear accident. According to a prior study, about half of the population believed that “offspring effects will occur due to the FDNPS accidents”, even though there is no scientific evidence to support this [28]. People who were in Fukushima at the time of the accident reported serious issues of discrimination regarding radiation and health effects [29]. This study has shown that avoiding harmful rumors about their children was a key motivator for self-evacuation. It is essential to increase public awareness and take the necessary steps to protect children’s dignity from the reputational damage associated with radiation.

This study has some limitations. The results were analyzed based on the KJ method, using the scientific data gathered from interviews with 27 women. However, these women did not all face the same situations or events during their self-evacuations, as each person’s experiences were unique. Furthermore, some benefits of continuing their self-evacuation, even in unfamiliar regions to the evacuees, were noted [30]. As such, it is important to interpret these results with caution. The study highlighted the challenges faced by mothers and small children during self-evacuation. The difficulties faced by mothers might differ between those with preschool-aged children and those with school-aged children. Previous research has reported that school-aged children who experienced the nuclear accident faced other challenges, such as bullying, due to the need to evacuate [31].

Despite these limitations, this study revealed and organized the negative experiences of self-evacuation had by mothers of small children following the FDNPS accident, which were previously unknown. Generally, both the women and children were vulnerable and easily began to struggle in their situation following the disaster [32,33]. The study indicated that these mothers who chose to evacuate to an unfamiliar region experienced highly stressful situations. If their views on evacuation differed from those of key persons in their family or social circle, the mother had a tougher experience in the unfamiliar region. The findings of this study have deepened our understanding of the hardships faced by indirect victims of the nuclear accident, such as women parenting small children.

## 5. Conclusions

This study clarified the negative aspects of self-imposed evacuation encountered by mothers of small children seeking to avoid radiation exposure from the FDNPS accident of 11 March 2011, using the Kawakita-Jiro (KJ) method. The results were classified into eight groups with 142 labels, including continued anxiety about the health effects of radiation, differences in risk perception, changes in spousal relationships, the inability to make friends and find support, living as a single parent, financial concerns, the unfamiliar feel of the area to which they evacuated, and uncertainty about the future. The study found that their anxiety about radiation exposure did not disappear even after evacuation. 

This research emphasizes the need to provide adaptable support to self-evacuees. Further, it is vitally important to educate people about radiation health effects, including improving their information literacy during the chaotic situations resulting from accidents and disasters.

## Figures and Tables

**Figure 1 ijerph-21-00592-f001:**
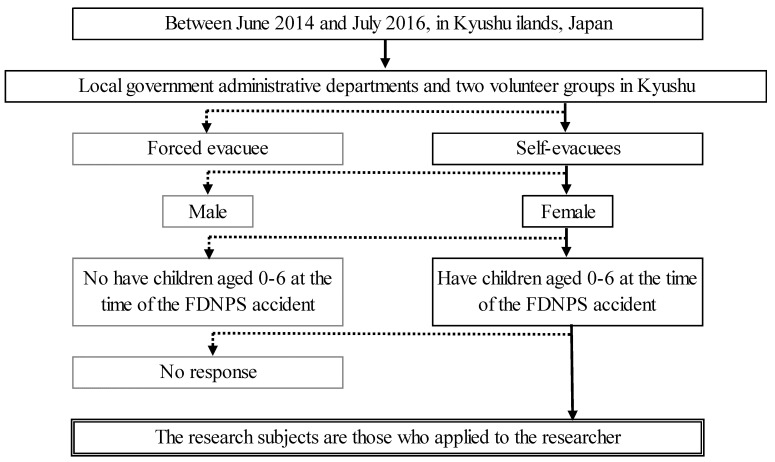
The data collection process in the study.

**Figure 2 ijerph-21-00592-f002:**
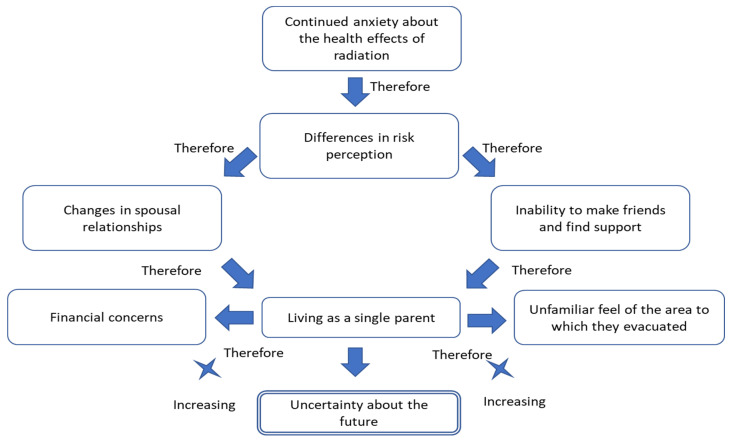
Difficulties of self-imposed evacuation among mothers of small children following Japan’s Fukushima Daiichi nuclear accident.

## Data Availability

Data available on request from the author.

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
