# Peer review of "Negative Aspects of Self-Imposed Evacuation among Mothers of Small Children Following Japan’s Fukushima Daiichi Nuclear Power Station Accident"

_ijerph, 2024, doi:10.3390/ijerph21050592_

Round 1
Reviewer 1 Report
Comments and Suggestions for Authors
The research highlights the impacts of nuclear power station mishap. Please find the following minor comments to enhance the quality of the manuscript prior to publication.
1. Abstract- why there a line gap in the abstract?
Please mention some of the hardships mentioned in abstract.
2. Introduction:- The introduction need to present the research gap as it is focused research.
What are the novelty aspects of this research?
3. The literature requires some more recent research citations.
4. How was the sample size of 27 mothers determined?
how were they selected? Sampling technique?
Any research philosophy or theoretical framework to support 27 respondents?
5. How was the story line dimensions selected?
6. Figure 1 is completed by the author or adapted from any source?
7. 3.2 any other views?
8. What about the role of the government? any initiatives from the private sector such as the local Small and Medium Enterprise (SMEs)?
9. Conclusion- please write two distinct paragraphs, one for conclusion and another for major recommendations from this research and conclude with future research implications.
Good Luck !!
Comments on the Quality of English LanguageEnglish proofread by a native speaker is suggested.
Author Response
Thank you for your valuable advice and feedback. As a result of your guidance, the following revisions have been implemented.
1. Abstract- why there a line gap in the abstract? Please mention some of the hardships mentioned in abstract.
→Thank you for bringing this to our attention. A line gap was our miss and fixed it (p1, line 11). Further, we added hardships within a 200-word maximum in the abstract (lines 17-18).
2. Introduction: - The introduction needs to present the research gap as it is focused research. What are the novelty aspects of this research?
According to your advice, I explained the novelty aspects of this research (p2,78-79).
3. The literature requires some more recent research citations.
According to your advice, I added some recent research citations (ref: 8,14,15).
4. How was the sample size of 27 mothers determined? how were they selected? Sampling technique? Any research philosophy or theoretical framework to support 27 respondents?
According to your suggestion, we have included the sampling technique and the subject selection process (p2,88-92). Nevertheless, we regret that there is no research philosophy or theoretical framework to support the participation of 27 respondents. We mentioned that “We found that after 27 interviews, most of the data points had already been discussed, so we declared that data saturation had been reached.” in the 2.2 data collection (p3,112-114).
5. How was the story line dimensions selected?
I apologize if my response is not suitable. The storyline dimension was selected by the most expressing of each categorized group among the low data. Further, I also refer to the previous studies when I analyzed and selected data. I added how to select the storyline dimensions in the results (p3,144-145).
6. Figure 1 is completed by the author or adapted from any source?
Thank you for pointing out the importance of the main results. The creation of Figure 1 was accomplished by the author (p3,128-130).
7. 3.2 any other views?
According to your advice, I have included "social isolation" as an additional perspective (p5, lines 183-184).
8. What about the role of the government? any initiatives from the private sector such as the local Small and Medium Enterprise (SMEs)?
According to your advice, I added the role of the private sector regarding the self-evacuation (p1, lines 62-64).
9. Conclusion- please write two distinct paragraphs, one for conclusion and another for major recommendations from this research and conclude with future research implications.
Based on your proposal, we have divided it into two separate paragraphs. Additionally, we have included potential future research implications in the conclusions (p9, lines 375-378).
Following your suggestion, I submitted the revised manuscript to an English editing service.
Reviewer 2 Report
Comments and Suggestions for Authors
This paper takes an in-depth look at the negative effects on mothers of young children who sought self-evacuation to avoid exposure to radiation during the 2011 Fukushima Daiichi nuclear power plant accident. By using semi-structured interviews with 27 mothers, the authors conducted data analysis using the Kawakita-Jiro (KJ) method to divide the difficulties associated with child self-evacuation into eight groups.
The paper's topic is clear, logical, and novel in that it focuses on a specific group of self-evacuees who are not supported by the government, rather than those who are officially ordered to evacuate, and uses the Kawakita-Jiro (KJ) method for data analysis to systematically classify and clarify the complex experiences of these mothers. It helps to understand the broader effects of a nuclear accident beyond the immediate physical health risks. This study lays the foundation for the development of support structures to address the multifaceted challenges faced by self-evacuees in future nuclear events. Suggested to publish after minor revision:
(1) Expanding the Literature Review: The introduction section could be strengthened by including a more comprehensive review of previous studies on self-imposed evacuation, particularly focusing on mental health outcomes and social integration challenges. This would provide a stronger foundation for the study and highlight the research gap the study aims to fill.
(2) Charts and related data materials are added to make it easier for readers to understand the research results and the relevant data collection process.
Author Response
Thank you for your valuable advice and feedback. As a result of your guidance, the following revisions have been implemented.
(1) Expanding the Literature Review: The introduction section could be strengthened by including a more comprehensive review of previous studies on self-imposed evacuation, particularly focusing on mental health outcomes and social integration challenges. This would provide a stronger foundation for the study and highlight the research gap the study aims to fill.
According to your suggestions, I included a comprehensive review of previous studies on self-imposed evacuation. (p2, line 71-79)
(2) Charts and related data materials are added to make it easier for readers to understand the research results and the relevant data collection process.
According to your suggestions, I added the charts and related data materials in Figure 1 (p3).
Reviewer 3 Report
Comments and Suggestions for Authors
In general, the topic raised in this research is very interesting. Some things that might be of note to the author are as follows:
1. In the introduction, the location of the Fukushima nuclear power has not been explained in detail
2. Should it be explained why this research is so urgent? Isn't there still a lot of other, more urgent research?
3. Provide a comprehensive explanation, what changes can educational institutions make to be able to implement disaster curricula, especially those related to nuclear?
4. This paper will be more interesting if it explains disaster mitigation and educational models that can be applied to students and can be integrated with existing topics in schools
5. What things can parents do to maximize disaster mitigation efforts that have been carried out at school?
Author Response
Thank you for your valuable advice and feedback. As a result of your guidance, the following revisions have been implemented.
In general, the topic raised in this research is very interesting. Some things that might be of note to the author are as follows:
1. In the introduction, the location of the Fukushima nuclear power has not been explained in detail.
According to your suggestions, we added the location of the FDNPS in the introduction (p1, line 28).
2. Should it be explained why this research is so urgent? Isn't there still a lot of other, more urgent research?
Thank you for your suggestion, we added why the research is urgent in p1, Lines 78-79. In my opinion, this research is more important than urgent.
3. Provide a comprehensive explanation, what changes can educational institutions make to be able to implement disaster curricula, especially those related to nuclear?
Thank you for the excellent advice. Following your guidance, I comprehensively suggested that educational institutions incorporate disaster curricula focusing on radiation into their programs (p8, lines 328-332).
4. This paper will be more interesting if it explains disaster mitigation and educational models that can be applied to students and can be integrated with existing topics in schools
Thank you for the excellent guidance. I followed your advice and suggested educational models about radiation in the existing school drills, specifically on p8, lines 332-334.
5. What things can parents do to maximize disaster mitigation efforts that have been carried out at school?
Thank you for the valuable advice. Consequently, I included information about the education of radiation for parents (page 8, lines 334-337).